# *Centella asiatica*-Derived Endothelial Paracrine Restores Epithelial Barrier Dysfunction in Radiation-Induced Enteritis

**DOI:** 10.3390/cells11162544

**Published:** 2022-08-16

**Authors:** Seo Young Kwak, Won Il Jang, Seung Bum Lee, Min-Jung Kim, Sunhoo Park, Sang Sik Cho, Hyewon Kim, Sun-Joo Lee, Sehwan Shim, Hyosun Jang

**Affiliations:** 1Laboratory of Radiation Exposure & Therapeutics, National Radiation Emergency Medical Center, Korea Institute of Radiological and Medical Sciences, Seoul 01812, Korea; 2Department of Surgery, Korea Institute of Radiological and Medical Sciences, Seoul 01812, Korea

**Keywords:** *Centella asiatica*, radiation-induced enteritis, epithelial barrier dysfunction, epidermal growth factor, endothelial paracrine

## Abstract

Radiation-induced enteritis is frequently observed following radiotherapy for cancer or occurs due to radiation exposure in a nuclear accident. The loss of the epithelial integrity leads to ‘leaky gut’, so recovery of damaged epithelium is an important strategy in therapeutic trials. *Centella asiatica* (CA), a traditional herbal medicine, is widely used for wound healing by protecting against endothelial damage. In this study, we investigated the radio-mitigating effect of CA, focusing on the crosstalk between endothelial and epithelial cells. CA treatment relieved radiation-induced endothelial dysfunction and mitigated radiation-induced enteritis. In particular, treatment of the conditioned media from CA-treated irradiated endothelial cells recovered radiation-induced epithelial barrier damage. We also determined that epidermal growth factor (EGF) is a critical factor secreted by CA-treated irradiated endothelial cells. Treatment with EGF effectively improved the radiation-induced epithelial barrier dysfunction. We also identified the therapeutic effects of CA-induced endothelial paracrine in a radiation-induced enteritis mouse model with epithelial barrier restoration. Otherwise, CA treatment did not show radioprotective effects on colorectal tumors in vivo. We showed therapeutic effects of CA on radiation-induced enteritis, with the recovery of endothelial and epithelial dysfunction. Thus, our findings suggest that CA is an effective radio-mitigator against radiation-induced enteritis.

## 1. Introduction

Radiation-induced enteritis is observed following clinical application of radiotherapy for pelvic cancer, and unexpected radiation exposure in a nuclear accident also leads to severe life-threatening intestinal injury. Severe intestinal damage, with inflammation, insufficient epithelial cell production, and instability [1,2], leads to side effects such as vomiting, weight loss, diarrhea, infections, and septic shock-induced death [2]. The endothelium has already been described as a crucial component involved in gastrointestinal (GI) diseases, such as radiation-induced enteritis and inflammatory bowel disease (IBD) [3,4], and it has been proposed that the pathogenesis of radiation-induced enteritis is associated with endothelial dysfunction [5,6,7]. Radiation exposure induces many changes in endothelial cells, such as apoptosis, senescence, increased endothelial permeability, interstitial fibrin deposition, and altered paracrine signaling. Because microvascular endothelial cells are located very close to the epithelial cells in the intestine, epithelial and endothelial cells can interact with each other by the release of growth factors and hormones. Communication from epithelial to endothelial cells is well-accepted [8,9,10,11]. In contrast, little is known about the respective backward communication from endothelial to epithelial cells.

The epithelial barrier is the line of physical defense in the GI tract that prevents the diffusion of pathogens into the intestinal mucosa. Barrier damage leads to increased epithelial permeability and bacterial translocation to internal organs, followed by leukocyte accumulation, inflammation, and sepsis. Thereby, the preservation of the epithelial integrity is a major aspect in order to preserve homeostasis and avoid the progress of inflammation in mucosal tissue [12]. Intercellular junctions of epithelial cells include adherens junctions (AJs), tight junctions (Claudin 3; Cldn3 and Zonula occludens 1; Zo1), and desmosomes (Desmoglein 2; Dsg2) and have a critical role in the epithelial integrity [13]. Uncontrolled epithelial permeability with decreased expression of intercellular junctions is a hallmark in radiation-induced enteritis and IBD patients [14,15]. Factors that prevent epithelial barrier dysfunction are important in developing therapeutic drugs for radiation-induced GI damage [16,17].

*Centella asiatica* (CA), known as Asiatic pennywort, is widely used as a traditional herbal medicine in China and India. This tropical medicinal plant is enriched with bioflavonoids, triterpenes, and selenium and has been reported to promote healing for ulceration, diarrhea, mental clarity, depression, and skin psoriasis [18,19,20,21]. In recent years, much attention has been paid to the potential of CA in the treatment of various types of disease, and some putative mechanisms have been proposed, including antioxidant and lipid metabolism in the skin and neurons [22,23]. Importantly, it has also been reported that CA can protect endothelial cells, increase cell proliferation, inhibit apoptosis of endothelial cells, and block beta-amyloid peptide aggregation [24]. Madecassoside, one of the triterpines isolated from CA, is known to preserve endothelial cells from oxidative injury by the protection of mitochondria membrane potential and apoptosis [25]. Asiatic acid, which is another component of CA, attenuates tumor necrosis factor-alpha-induced endothelial barrier dysfunction, thereby resulting in the prevention of atherosclerosis [26]. However, the effect of CA on radiation-induced endothelial cell damage has not yet been investigated.

The results of the present study show that CA ameliorated radiation-induced enteritis with the recovery of endothelial cell damage and epithelial barrier dysfunction. We hypothesized that the soluble factor secreted by CA-treated irradiated endothelial cells could repair radiation-induced enteritis by regulating the epithelial barrier. We found that the conditioned media (CM) of CA-treated irradiated endothelial cells reversed the radiation-induced epithelial barrier dysfunction in vitro as well as in radiation-induced enteritis in a mouse model. We also discovered that CA treatment of irradiated endothelial cells induced the secretion of epidermal growth factor (EGF), which is necessary for the repair of radiation-induced epithelial barrier dysfunction, including integrity and expression of junction proteins. Of particular note, blocking EGF in CM using a neutralizing antibody failed to rescue the epithelial barrier dysfunction. We identified the therapeutic effects of CA-induced endothelial secretome in a radiation-induced enteritis mouse model with epithelial barrier restoration. Otherwise, CA treatment did not occur radioprotective effects on colorectal tumors in vivo.

Collectively, the results indicated that CA recovered radiation-induced epithelial damage by regulating endothelial-derived EGF and that the interaction of epithelial-endothelial is a novel therapeutic target that can be used to alleviate radiation-induced enteritis.

## 2. Materials and Methods

### 2.1. Cell Culture and Reagents

Human umbilical vein endothelial cells (HUVECs; Lonza, Basel, Switzerland) were cultured in an EGM-2 medium supplemented with endothelial growth kit components (Lonza). The passage number of HUVECs used in experiments was between 4 and 7. Human Caco-2 cells were maintained in Dulbecco’s Modified Eagle Medium (DMEM, Gibco, Grand Island, NY, USA) supplemented with 10% fetal bovine serum (FBS, Gibco) and 1% antibiotics. The human colon cancer cell line (HCT-116) was maintained in RPMI medium (Gibco) with 10% FBS and 1% antibiotics. All cells were grown in a humidified incubator at 37 °C with 5% humidity. Based on previous studies [27,28], Caco-2 cells were grown into a confluent monolayer for in vitro experiments as a barrier function model. To obtain CM from HUVECs, irradiated HUVECs were either treated or not with CA (US Pahrmacopeia, Rockville, MD, USA) for 24 h. After incubation, the media was exchanged with fresh serum-free EBM-2 media. Caco2 used in this study was between passages 18 and 33.

### 2.2. Animals

Specific pathogen-free male C57BL/6 mice were obtained from Harlan Laboratories (Indianapolis, IN, USA) and maintained in specific pathogen-free conditions at the Korea Institute of Radiological & Medical Sciences (KIRAMS) animal facility. All mice were housed in a temperature-controlled room with a 12-h light/dark cycle. Food and water were provided *ad libitum*. The mice were acclimated for 1 week before the commencement of the experiments and were grouped as follows: control (Con), irradiation (IR), irradiation with CA treatment (IR + CA), irradiation with CM from irradiated HUVECs treatment (IR + CM), irradiation with CM from CA-applied irradiated HUVECs treatment (IR + CA-CM), and irradiation with recombinant EGF (rEGF) treatment (IR + rEGF). All animal experiments were approved and performed in accordance with the guidelines of the Institutional Animal Care and Use Committee of the KIRAMS (kirams 2020-0010).

### 2.3. Irradiation and Treatment

Cells were irradiated to 10 or 15 Gy using a ^137^Cs γ-ray source (Atomic Energy of Canada, Ltd., Laurentian Hills, ON, Canada) with a dose rate of 3.25 Gy/min. Animals were anesthetized with 85 mg/kg alfaxalone (Alfaxan^®^, Careside, Gyeonggi-do, Korea) and 10 mg/kg xylazine (Rompun^®^ Bayer Korea, Seoul, Korea). Mice were irradiated in the abdomen with a single dose at 13.5 Gy using an X-RAD 320 X-ray irradiator (Softex, Gyeonggi-do, Korea). Within 2 h after exposure to radiation, CA (200 µg/kg/day), the CM of irradiated HUVECs, the CM of CA-treated irradiated HUVECs (200 µL/mouse/day), and rEGF (1 µg/mouse/day) were administrated, and these treatments were continued for 6 days.

### 2.4. CCK-8 Assay

HUVECs were seeded in a 96-well plate. On the next day, cells were irradiated at 10 Gy and treated with varying concentrations of CA. After a 48-hour incubation, the CCK-8 reagent was added and measured using a microplate reader at a wavelength of 450 nm. The experiments were carried out at least in triplicate.

### 2.5. β-Galactosidase Assay

HUVECs were irradiated at 10 Gy using a ^137^Cs γ-ray source (Atomic Energy of Canada, Ltd., Laurentian Hills, ON, Canada) with a dose rate of 3.25 Gy/min. Irradiated HUVECs were subsequently treated with CA for 48 h. Prepared cells were fixed with 4% paraformaldehyde and subsequently stained using a b-galactosidase kit (Cell Signaling Technology, Danvers, MA, USA) according to the manufacturer’s instructions.

### 2.6. Tube Formation Assay

Irradiated HUVECs were re-seeded onto Matrigel-coated transwell (Corning, NY, USA), followed by treatment with or without CA for 6 h. Total tube length was observed under a light microscope and plotted using Image J.

### 2.7. Histological Analysis of the Intestine

Mouse small intestinal tissue samples were fixed with a 10% neutral buffered formalin solution, embedded in paraffin wax, and sectioned transversely at a thickness of 4 µm. The sections were then stained with hematoxylin and eosin (H&E). The length of 15 villi and the number of crypts per circumference present in at least four cross-sections per mouse were analyzed. The severity of radiation-induced enteritis was assessed by the degree of maintenance of the epithelial architecture, crypt damage, vascular enlargement, and infiltration of inflammatory cells in the lamina propria. This assessment is a modification of the histological score parameter used by Sung et al. [29]. To perform immunohistochemical analysis, slides were subjected to antigen retrieval and then treated with 0.3% hydrogen peroxide in methyl alcohol for 20 min to block endogenous peroxidase activity. After three washes in PBS, the sections were blocked with 10% normal goat serum (Vector ABC Elite kit; Vector Laboratories, Burlingame, CA, USA) and incubated with anti-Zo1 (#61-7300, Thermo Fisher Scientific, Waltham, MA, USA), anti- Dsg2 (#14415, Abcam), and anti- Cldn3 (#341700 Invitrogen, Carlsbad, CA, USA), anti-villin (#130751, Abcam), anti-ki-67 (#DRM004, Acris, Herford, Germany), anti-Cd68 (#125212, Abcam), anti-Cd31 (#28364, Abcam), anti-Cd34 (#SC-74499, Santa-Cruz), and anti-Olfm4 (#39141, Cell Signaling) antibodies. After three washes in PBS, the sections were incubated with a horseradish peroxidase-conjugated secondary antibody (Dako, Carpinteria, CA, USA) for 60 min. The peroxidase reaction was developed using a diaminobenzidine substrate (Dako) prepared according to the manufacturer’s instructions, and the slides were counterstained with hematoxylin.

### 2.8. Immunocytochemical Staining

Caco-2 monolayers on coverslips were harvested, and immunofluorescence analysis was performed. Cells were fixed with paraformaldehyde, blocked and permeabilized with 1% BSA and triton-X100 for 30 min at room temperature, and incubated with the primary antibodies specific for ZO1 and DSG2. Samples were incubated for 1 h at room temperature with the Alexa Fluor 488 (green)-conjugated anti-rabbit IgG and Alexa Fluor 592 (red)-conjugated anti-mouse IgG (Thermo Fisher Scientific) as secondary antibodies. After washing with PBS, cells were count-stained with DAPI and mounted using Vectashield HardSet mounting medium. Fluorescence was examined using a confocal laser scanning microscope (LSM410; Carl Zeiss, Oberkochen, Germany).

### 2.9. Bacterial Translocation

To evaluate barrier function, treated mice were sacrificed, and the mesenteric lymph nodes were harvested under sterile conditions. The mesenteric lymph nodes were homogenized with sterile PBS and beads. The homogenized mixtures were centrifuged to remove cell debris and subsequently spread onto MacConkey agar (BD Biosciences). After incubation overnight, the colony-positive plates were counted. Data were graphed as the percentage of individual mice exhibiting colonies compared to individual control mice.

### 2.10. RNA Extraction and qPCR

Total RNA of cells and in vivo samples was extracted using Tri-reagent (MRC, Cincinnati, OH, USA) according to the manufacturer’s instructions. cDNA was synthesized using the AccuPower RT premix (Bioneer, Deajeon, Korea). Synthesized cDNA was amplified using a LightCycler 480 system (Roche, Basel, Switzerland) with specific primers. Expression levels of each gene were determined using the Delta-Delta-Ct (ddCt) method. The sequences of the primers were as follow: mouse *villin*, 5′-CACCTTTGGAAGCTTCTTCG-3′ (forward) and 5′-CTCTCGTTGCCTTGAACCTC-3′; mouse *tight junction protein 1* (*Tjp1)*, 5′-AGGACACCAAAGCATGTGAG-3′ (forward) and 5′- GGCATTCCTGCTGGTTACA-3′; mouse *Dsg2*, 5′-GTAGGAGGTGCGATGCTCAA-3′ (forward) and 5′- CATGCTGCCCTTTGGTAACG-3′ (Reverse); mouse *Cldn3,* 5′-AAGCCGAATGGACAAAGAA-3′ (forward) and 5′-CTGGCAAGTAGCTGCAGTG-3′ (Reverse); mouse *Il1b*, 5′- GCAACTGTTCCTGAACTCA-3′ (forward) and 5′- CTCGGAGCCTGTAGTGCAG-3′ (Reverse); mouse *Mcp1*, 5′- AGGTCCCTGTCATGCTTCT-3′ (forward) and 5′- CTGCTGGTGATCCTCTTGT-3′ (Reverse); mouse *Lgr5*, 5′-TCAGTCAGCTGCTCCCGAAT-3′ (forward) and 5′- CGTTTCCCGCAAGACGTAAC-3′ (Reverse); mouse *Olfm4*, 5′-GCTGGAAGTGAAGGAGATGC-3′ (forward) and 5′- ACAGAAGGAGCGCTGATGTT-3′ (Reverse); mouse *Gapdh,* 5′-TCCCTGGAGAAGAGCTATGA-3′ (forward) and 5′-CGATAAAGGAAGGCTGGAA-3′ (reverse); human *ZO1,* 5′-ATCCCTCAAGGAGCCATTC-3′ (forward) and 5′- CACTTGTTTTGCCAGGTTTTA-3′ (reverse); human *DSG2,* 5′-TGGACACCCAAACAGTGGCCCT-3′ (forward) and 5′- CTCACTTTGTTGCAGCAGCACAC-3′ (reverse); human *EGF,* 5′-GTGCAGCTTCAGGACCACAA-3′ (forward) and 5′- AAATGCATGTGTCGAATATCTTGAG-3′ (reverse); human *GAPDH,* 5′-GGACTCATGACCACAGTCCATGCC-3′ (forward) and 5′-TCAGGGATGACCTTGCCCACAG-3′ (reverse).

### 2.11. Western Blot

Cell lysates were washed with PBS and lysed in cold RIPA supplemented with a cocktail of protease and phosphatase inhibitor (Roche) on ice. Protein concentrations were determined by a bicinchoninic acid (BCA) method using Pierce BCA protein Assay (Thermo Fisher Scientific). An equal quantity of samples mixed with sodium dodecyl sulfate (SDS)-containing sample buffer were boiled at 95 °C for 5 min and separated by SDS-poly acrylamide gel electrophoresis. Proteins were transferred to polyvinylidene fluoride for immunoblotting (Bio-rad, Hercules, CA, USA). The membrane was blocked with 5% skim milk in TBS. Primary antibodies diluted in tris-buffered saline and Tween 20 (TBS-T) were incubated overnight at 4 °C. The following antibodies were used: anti-ZO1 (Thermo Fisher Scientific), anti-DSG2 (Abcam), and anti-β-Actin (Santa Cruz, CA, USA). Following overnight incubation, the membrane was washed and incubated with horseradish peroxidase-conjugated secondary antibodies for 1 h (Santa Cruz) diluted in TBS-T. The membrane was washed, and proteins were detected using an enhanced chemiluminescence reagent (Pierce, Thermo Fisher Scientific).

### 2.12. Transepithelial Electrical Resistance (TEER) Measurement

Caco-2 cells were seeded into the upper chamber of the transwell (0.4 µm pore size, Corning) and cultured for 21 days to form epithelial monolayers. Caco-2 monolayers were exposed to radiation and followed by treatment with various experimental conditions. The EVOM system (WPI, Sarasota, FL, USA) was used to measure TEER values.

### 2.13. Fluorescein Isothiocyanate (FITC)-Dextran Flux Measurement

Caco-2 cells were seeded into the upper chamber of transwell inserts (0.4 µm pore size, Corning) and cultured for 21 days. Caco-2 monolayers in the transwell were irradiated and incubated under various experimental conditions with 500 µg/mL of FITC-dextran (Sigma-Aldrich, St. Louis, MO, USA). Media in the lower chamber were taken after 48 h, and fluorescence was subsequently measured using a microplate fluorescence reader (excitation at 450 nm and emission at 520 nm). The flux of FITC into the lower chamber was calculated as a percentage corresponding to the control sample.

### 2.14. Dispase-Based Dissociation Assay

To evaluate cell–cell adhesive strength, Caco-2 monolayers were washed and incubated in dispase II (2.4 U/mL, Roche) and collagenase type I (Gibco) for 30 min and were released from the well bottom. To apply mechanical stress, the Caco-2 monolayers were carefully subjected to pipetting 5 times with an automatic pipet. The attached cell sheets were observed by a digital camera.

### 2.15. Human Protein Cytokine Array

HUVECs were irradiated and followed by CA or not in complete media. After 24 h, cells were washed once with PBS and exchanged for a fresh serum-free medium. The CM of HUVECs were collected and spun down to remove cell debris. The CM was analyzed using the proteome profiler^TM^ Human Cytokine Array Kit (R&D systems, Minneapolis, MN, USA) according to the manufacturer’s instructions. Densitometry was performed with Image J (National Institute Health) to determine the relative abundance of cytokines in the CM.

### 2.16. Enzyme-Linked Immunosorbent Assay (ELISA)

To quantify EGF, the CM was collected and spun down to remove cell debris. The CM was subjected to ELISA (R&D Systems) according to the manufacturer’s instructions.

### 2.17. Neutralization of EGF

In the neutralizing experiment, each CM sample was prepared as described above and incubated with 100 ng/mL of anti-EGF (R&D systems) for 1 h to bind the antibody. Caco-2 monolayers were washed with PBS, and pre-incubated medium was added. Cells were incubated for 48 h and subsequently analyzed by additional assays.

### 2.18. In Vivo Tumor Growth Assay

To determine the effect of treatment with CA on irradiated tumor growth in vivo, a xenograft model was performed in Balb/c nude mice (Orient-Bio Laboratory, Korea, Seoul). Balb/c nude mice were injected subcutaneously with 1 × 10^7^ HCT116 cells on right hinge. When the tumor size reached a mean volume of 100 mm^2^, mice were randomly divided into three groups (*n* = 7/group): (1) Con, mice did not receive any treatment, (2) IR alone, mice were exposed to IR at a dose of 2 Gy at daily for 5 days; (3) IR + CA, mice were irradiated and subsequently administered an intraperitoneal injection with CA (200 mg/kg body weight). Tumors were measured in two dimensions with calipers every 3 days. Tumor volumes were calculated using the formula: V(mm^3^) = length (mm) × width^2^ (mm^2^) × 1/π. Mice were sacrificed when tumor mass was reached at 1000 mm^3^. All animal experiments were approved and performed in accordance with the guidelines of the Institutional Animal Care and Use Committee of the KIRAMS (kirams 2022-0002).

### 2.19. Statistical Analysis

The in vitro data were plotted as mean ± standard deviation of the mean, and animal data were plotted as the mean ± standard error of the mean. Statistical analyses were performed using one-way analysis of variance (ANOVA) with Tukey’s multiple comparison test. Values of *p* < 0.05 were considered statistically significant.

## 3. Results

### 3.1. CA-Treated Endothelial Cells Are Recovered from Radiation-Induced Dysfunction and Protects Epithelial Barrier Damage

To investigate the therapeutic effects of CA on irradiated endothelial cells, we performed several assays using HUVECs in the presence or absence of CA. We used the CCK-8 assay in irradiated HUVECs to assess cell viability. Irradiation of HUVECs showed significant downregulation of cell viability compared to the control, but CA treatment rescued the radiation-induced loss of cell viability (Figure 1A,B). Because radiation induces cellular senescence [30], we tested the cellular senescence activity using a β-galactosidase assay. The β-galactosidase activity was observed in irradiated HUVECs, but CA treatment of irradiated HUVECs displayed lower β-galactosidase activity than irradiated HUVECs (Figure 1C). A tube formation assay was performed to assess angiogenic capacity. The tube-forming activity of HUVECs was inhibited by radiation, but CA treatment restored the angiogenic activity of irradiated HUVECs (Figure 1D). These results suggest that CA mitigated radiation-induced endothelial dysfunction, including viability, senescence, and angiogenic properties.

The epithelial barrier requires a monolayer of epithelial cells to separate organs from the extracellular environment. An intact epithelium, which is considered to be responsible for protection against exogenous pathogens, is constantly exposed to soluble factors produced by surrounding cells in the microenvironment [8,9,10,11]. Especially, interactions between endothelial and epithelial cells play important roles in controlling intestinal barrier function under pathological conditions [8,9]. Considering CA improved endothelial cell function in vitro on radiation exposure, it was decided to evaluate the functional effect of a CA-treated endothelial paracrine on epithelial cell damage repair. We used well-established in vitro models reflecting epithelial barriers [28,31] to evaluate the functional activity of endothelial paracrine on a damaged epithelial barrier. The CM of HUVECs was collected after irradiation (CM of IR) or IR followed by CA treatment (CM of IR + CA) in a serum-free medium. The CM of each sample was tested on a Caco-2 monolayer. As shown in Figure 1D, the TEER value of the CM of IR-treated IR Caco-2 monolayers was decreased compared with that of CM of IR-treated non-IR Caco-2 monolayers. Otherwise, the CM of IR + CA treatment increased the TEER value in IR Caco-2 monolayers (Figure 1E). In addition, the FITC flux of the CM of IR-treated IR Caco-2 monolayers was increased compared to the CM of IR-treated non-IR Caco-2 monolayers. The CM of IR + CA treatment on IR Caco-2 monolayers decreased FITC flux in the FITC-dextran assay (Figure 1F). The cell–cell contact strength of IR Caco-2 monolayers was also improved by the CM of IR + CA treatment (Figure 1G). We used immunofluorescence to evaluate the expression of barrier integrity-related molecules. While ZO1 and DSG2 were lost in the junctions of the CM of IR-treated IR Caco-2 monolayers, the loss of junctional molecules was recovered by CM of IR + CA treatment of IR Caco-2 monolayers (Figure 1H). Consistent with these results, protein and mRNA levels of ZO1 and DSG2 were decreased in the CM of IR-treated IR Caco-2 monolayers, whereas these expressions were restored by treatment of the CM of IR + CA (Figure 1I,J). Collectively, CA modulated the endothelial paracrine to restore radiation-induced barrier dysfunction, particularly that associated with ZO1 and DSG2.

### 3.2. CA Mitigates Radiation-Induced Enteritis and Improves Intestinal Barrier Dysfunction in Mouse Model

We evaluated the therapeutic effects of CA in a radiation-induced enteritis mouse model in which the abdomen of the mouse was irradiated. Mice were then either treated with CA (IR + CA) or left untreated (IR). Six days after irradiation, the effect of CA on radiation-induced enteritis was determined using physiological and histological examinations. CA administration to the IR mouse attenuated loss of body weight compared to the IR group (Figure 2A). Histological analyses of irradiated intestine showed shorter villi length and crypt disruption, whereas CA treatment restored villi length and crypt numbers in IR mice (Figure 2C). Histological scoring, accomplished by evaluating epithelial structural damage, vascular dilation, and inflammatory cell infiltration in the mucosa and submucosa, was lower in IR + CA mice than in the irradiated group (Figure 2B). Immunohistochemical activity for Ki-67, a proliferation marker, was also increased in the IR + CA group compared to the IR group (Figure 2D). Radiation-induced enteropathy is characterized by an inflammatory response with increased inflammatory cytokines such as IL1β and MCP1 and inflammatory cell infiltration. CA treatment inhibited the inflammatory cytokines and Cd68-expressed monocyte infiltration in the irradiated intestine (Figure 2E,G). Lgr5 and Olfm4 are specific markers of intestinal stem cells. We observed that these markers markedly decreased in the IR group compared with control. CA treatment significantly upregulated these markers in IR conditions (Figure 2F,G). As indicated by immunohistochemistry for the endothelial cell marker CD31 and endothelial progenitor cell marker CD34, angiogenic continuity and endothelial progenitor cells were also increased in IR + CA mice than in the IR group (Figure 2H,I). Taken together, these results suggest that CA inhibits inflammation and recovers radiation-induced enteritis with the restoration of endothelial dysfunction.

Next, we investigated whether CA affects radiation-induced intestinal barrier dysfunction in a mouse model system. We evaluated bacterial translocation in mesenteric lymph nodes as a measure of the intestinal barrier. The bacterial translocation in the mesenteric lymph nodes was increased in the IR group compared to the control group, but it was decreased in the IR + CA group compared to the IR group (Figure 3A). Next, we assessed expressions of the intercellular junction molecules regulating the barrier function. Immunohistochemistry analysis showed that positive cells for epithelial barrier-related molecules, such as villin, Zo1, Dsg2, and Cldn3, were decreased in the IR group compared to the control group. However, these expressions were restored in the IR + CA group (Figure 3B). We also assessed the mRNA levels of these molecules in intestinal tissue. The mRNA levels of epithelial barrier-related molecules in the IR group showed a significantly lower expression compared to the control group, but CA treatment restored mRNA expression (Figure 3C). Taken together, these results suggest that CA attenuated radiation-induced intestinal enteritis thereby avoiding intestinal barrier dysfunction in a mouse model.

### 3.3. CA Accelerates EGF Production in Irradiated Endothelial Cells

Endothelial cells communicated with epithelial cells by secreting a variety of biologically active growth factors, cytokine, extracellular matrix protein, and tissue remodeling enzymes [32]. The paracrine factors from endothelial cells may help restore the GI epithelium. To elucidate which secretory molecules influence the repair of radiation-induced epithelial dysfunction, we performed cytokine array experiments to analyze secretome profiling. Each CM of HUVECs (i.e., control (Con), irradiated HUVECs (IR), CA-treated HUVECs (CA), and CA-treated irradiated HUVECs (IR + CA)) was applied to the cytokine array. Cytokine analysis revealed changes in several factors, including EGF, interleukin (IL)-6, and IL-8 (Figure 4A). While IL-6 and IL-8 levels significantly decreased in the IR + CA group compared to the IR group, there was no response to irradiation. Interestingly, the EGF level decreased in the IR group compared with Con group and was significantly upregulated in the IR + CA group compared with the IR group. EGF, a well-known growth factor, plays a critical role in cell proliferation and protects the GI mucosa from a variety of insults [33,34]. The paracrine levels of EGF, determined using ELISA methods, were significantly increased 2-fold in the IR + CA group compared to the IR group (Figure 4B). EGF-positive HUVECs increased in immunofluorescence in the IR + CA group compared to the IR group (Figure 4D). The mRNA level of *EGF* was decreased in the IR group, but its expression was recovered in the IR + CA group (Figure 4C). These results suggest that CA treatment of irradiated endothelial cells induced EGF production and secretion.

### 3.4. EGF Restores Radiation-Induced Epithelium Barrier Dysfunction with Upregulation of ZO1 and DSG2

To determine whether CA-induced endothelial EGF secretion could ameliorate radiation-induced epithelium barrier dysfunction, we performed epithelial barrier functional assays using recombinant EGF (rEGF).

As shown in Figure 5A, the decreased TEER value in IR Caco-2 monolayers was increased by rEGF treatment (Figure 5A). An FITC-dextran assay indicated that FITC flux was elevated in media of IR Caco-2 monolayers, but it was diminished by rEGF treatment (Figure 5B). Cell–cell contact strength was decreased in IR Caco-2 monolayers but enhanced when rEGF was exposed to IR Caco-2 monolayers (Figure 5C). Confocal staining revealed that immunohistochemical activities against ZO1 and DSG2 were diminished in the intercellular junctions of IR Caco-2 monolayers but were reinforced by rEGF treatment (Figure 5D). The protein and mRNA levels of ZO1 and DSG2 had the same pattern as the confocal staining result (Figure 5E,F). EGF treatment of endothelial cells was also tested due to the possibility of an autocrine mode. The results indicate that rEGF treatment to HUVECs did not induce any mitigating effects such as viability, anti-senescence, and angiogenic ability (Supplement Appendix A–C). These results indicate that CA-derived EGF reverts radiation-induced epithelial barrier dysfunction, not radiation-induced endothelial damage.

### 3.5. CA-Derived Endothelial EGF Rescues Radiation-Induced Barrier Impairment with Upregulation of ZO1 and DSG2

To identify whether barrier function restoration by CA-derived endothelial paracrine is dependent on EGF, we abolished EGF in the CM of IR + CA using a neutralizing antibody (anti-EGF). As shown in Figure 6A, CM of IR + CA-treated with anti-EGF group significantly reduced the TEER value compared with CM of IR + CA group (Figure 6A). Blocking of EGF also failed to decrease the FITC flux of CM of IR + CA-treated IR Caco-2 monolayers (Figure 6B). A dispase-based dissociation assay showed that reinforcement of cell–cell contact strength in the CM group of IR + CA was abolished by neutralizing EGF (Figure 6C). Similarly, expression of epithelial barrier-related molecules in the CM of IR + CA-treated with the anti-EGF group did not increase as much as in the CM of the IR + CA group (Figure 6D,E). Upregulated mRNA levels of *ZO1* and *DSG2* in the CM of the IR + CA group were also abolished by anti-EGF treatment (Figure 6F). These results indicate that CA-derived endothelial EGF rescues radiation-induced epithelial barrier impairment with upregulation of ZO1 and DSG2.

### 3.6. CA-Derived Endothelial EGF Mitigates Radiation-Induced Enteritis with Epithelial Barrier Restoration in Mouse Model

To evaluate the therapeutic effect of CA-derived endothelial EGF on radiation-induced enteritis, we administered the CM of IR + CA HUVECs to an irradiated mouse model. The mouse groups were as follows: control (Con), irradiated (IR), irradiated and injected with the CM of IR HUVECs (IR + CM), irradiated and injected with the CM of IR + CA HUVECs (IR + CA-CM), and irradiated and injected with rEGF (IR + rEGF). Histological examination revealed that villi shortening and crypt disruption by radiation were rescued in the IR + CA-CM groups. Elevated histological scoring in the IR group was significantly reduced in the IR + CA-CM and IR + rEGF groups (Figure 7A,C). Otherwise, there were no significant differences in the IR and IR + CM groups (Figure 7A,C). Immunoreactivity for Ki-67 as a proliferating marker was also increased in the IR + CA-CM and IR + rEGF groups than in the IR + CM group and IR group (Figure 7B). Physiological examination showed that the body weight of the IR + CA-CM group was higher than that of the IR group on days 5 and 6 following treatment (Figure 7D). The inflammatory cytokines and inflammatory cells infiltration were significantly inhibited in the IR + CA-CM and IR + rEGF groups compared to the IR group (Appendix A). We observed that stem cell markers (Lgr5, Olfm4) markedly increased in IR + CA-CM group and IR + rEGF group compared to the IR groups (Appendix A). Of particular note, the immunohistochemical activity of Villin, Zo1, Dsg2, and Cldn3 increased in the IR + CA-CM and IR + rEGF groups compared to the IR group (Figure 7E). The mRNA levels, including *Villin*, *Zo1*, *Dsg2*, and *Cldn3*, in intestinal tissue, were also increased in the IR + CA-CM and IR + rEGF groups compared to the IR group (Figure 7F). These results suggest that CA-induced endothelial EGF efficiently alleviates radiation-induced enteritis and rescues barrier dysfunction.

### 3.7. CA Treatment Does Not Occur the Radioprotective Effect on Colorectal Tumor Growth

To investigate whether CA can be used as combination agents in tumoricidal radiotherapy, we treated CA in an irradiated HCT-116 (human colorectal cancer cell lines) xenograft mouse model. As shown in Figure 8A, the timeline of CA treatment and fractionated irradiation is illustrated (Figure 8A). Tumor-bearing mice were exposed to radiation (2 Gy/fractioned) for 5 days, followed by an intraperitoneal injection of CA at 200 mg/kg daily. IR group showed significant tumor growth retardation compared with the non-irradiated control group (Figure 8B,C), and the ratio of tumor mass per body weight was also decreased in the IR group (Figure 8D). Otherwise, mice receiving fractionated radiation with CA treatment (IR + CA) did not show a significant difference in tumor growth and mass compared with the IR group (Figure 8B–D). Body weights were not different in either group, indicating that the administration of CA did not cause systemic toxicity (Figure 8E). These results suggest that CA has no protective effect on colorectal tumor growth during radiation therapy.

## 4. Discussion

Radiotherapy is currently used as an indispensable therapy for a wide range of malignant conditions [35]. Radiation exposure for abdominopelvic cancers with varying degrees of GI tract complications, such as abdominal pain, loss of appetite, nausea, and diarrhea, which impede radiotherapy prematurely and worsen the life quality of patients [1]. It is estimated that millions of cancer survivors worldwide are suffering from intestinal dysfunction as a result of their cancer treatment [36], and the event of nuclear accidents or radiological terrorism is a significant source of morbidity and mortality by accelerating intestinal damage, such as inflammation, bacterial translocation and sepsis [2]. Therefore, radiation-induced intestinal injury is required development of therapeutic reagents, such as radiation-protector or radio-mitigator. Despite advances in radio-protectors (e.g., amifostine for acute radiation syndrome), there are no promising agents for an effective radio-mitigator for GI damage.

The potential medicinal plant CA is widely used in traditional medicine in the Orient and has been applied to skin lesions, ulcerations, and diarrhea [20,21]. In addition, its active constituents, primarily the main chemical components of pentacyclic triterpene derivatives (e.g., asiaticoside, asiatic acid, madecassoside, and madecassic acid), have been reported to recover the damaged tissue [37]. Madecassoside has been reported to protect endothelial cells against oxidative stress [25], and asiaticoside has been reported to heal the incision through the formation of an epithelial layer [38]. These reports indicate that CA is a promising reagent for the rescue of damaged tissues.

Treatment of CA as a radioprotector at a sublethal dose of Co-60 gamma irradiation has been shown to prolong the survival rate in a whole body irradiated mouse model [39]. Administration of CA has a protective effect on radiation-induced body weight loss and conditioned taste aversion [40]. However, no studies have been reported on the effect of CA on radiation-induced enteritis. In the present study, we investigated the radio-mitigating effect of CA, focusing on the crosstalk between endothelial and epithelial cells in vitro and in a mouse model. Because CA did not effectively improve in the epithelial cell line on high dose irradiation condition (data not shown), we found that CA ameliorates radiation-induced enteritis through modulation of radiation-induced endothelial cell paracrine. We also identified EGF as an endothelial-derived key regulator to repair radiation-induced epithelium disruption. Our findings also demonstrate that endothelial-derived EGF by CA treatment improved the epithelial barrier damage on radiation-induced enteritis.

Interactions between intestinal epithelial cells and the subepithelial cellular components play important roles in controlling intestinal barrier function under pathological conditions [8,9]. Studies have shown that crosstalk between endothelia and epithelial barrier is critical for regulation of tissue homeostasis and protection against pathogens or tissue-damaging agents in human airways [41]. The endothelial-epithelial paracrinal communication was studied using a human intestinal crypt cell line grown in noncontact co-culture with HUVEC. Endothelial cells secreted the 6-keto-prostaglandin F 1 alpha, a stable hydrolysis product of prostacyclin, resulting in epithelial cell activation through paracrine action [8]. Endothelial cells-derived Jagged 1 activates Notch in human colorectal cancer cells and thereby promotes a cancer stem cell phenotype and chemo-resistance [10]. Blockade of endothelial-induced CXCL10 enhances intestinal crypt cell survival in colitis model [11]. In this study, CM of CA-treated endothelial cells restored irradiated epithelial barrier damage. Therefore, we demonstrated that the secretome of CA-treated endothelial cells could rescue the radiation-induced epithelial dysfunction.

EGF, a well-known monomeric peptide present in the GI lumen, plays an important role in mitogenesis in tissue [42,43,44]. EGF and its related peptides have been implicated in the promotion of cell proliferation in wound healing, such as in re-epithelialization [45,46]. Furthermore, secreted EGF from bone marrow endothelial cells accelerates hematopoietic stem cell recovery [47]. It is well known that EGF treatment promotes survival after radiation exposure [48] and protects against radiation-induced intestinal injury [49]. Otherwise, there is little information about the effects of EGF on radiation-induced epithelial barrier damage. In our recent study, CA-derived EGF secretion rescued the impaired epithelial barrier in irradiated Caco-2 monolayers and in radiation damage mouse model. The use of EGF neutralizing antibody failed to rescue epithelial barrier dysfunction with upregulation of Zo1 and DSG2 of CA-derived endothelial paracrine. Taken together, these findings indicated that CA-derived endothelial EGF was a modulator that contributed to recovering the radiation-induced epithelial dysfunction. This is the first evidence of the functional cellular response of CA on damaged tissue.

Breakage of the epithelium barrier integrity is one of the important characteristics of radiation-induced enteritis. Gut epithelial barrier is the first defense to protect the extra insults. It has been reported that the epithelial barrier damaged by radiation or inflammatory stimuli leads to downregulation of TEER and integrity and fragmentation of cell–cell interactions [17,50]. Complexes of intercellular junctions, including tight junctions (e.g., ZO1, CLDN3), adherent junctions, and desmosomes (e.g., DSG2), are the principal components of the intestinal barrier. In particular, ZO1 alteration contributes to the disturbance of the epithelial barrier. Loss of ZO1 with barrier dysfunction has been shown in dextran sulfate sodium (DSS)-induced colitis and sepsis in a *pseudomonas aeruginosa* infection mouse model [51,52]. Epithelial ZO1-deficient mice display severe mucosal damage with increased permeability following DSS application [53]. Additionally, DSG2 is required for the integrity of the intestinal epithelial barrier in vitro and in vivo [54,55]. Intestinal epithelial DSG2 knockout mice exhibit severe colitis from DSS treatment with increased intestinal permeability [55]. In this study, CA-derived endothelial EGF increased the expression of ZO1 and DSG2 in irradiated Caco-2 monolayers and intestinal epithelium of radiation-induced enteritis. Taken together, upregulation of ZO1 and DSG2 by CA-induced EGF contributes to the recovery of epithelial barrier damage in irradiation.

CA components, such as asiatic acid and asiaticoside, inhibit the progression of cancer in the lung, colon, and skin [56,57,58]. However, the effect of CA on the progression of tumor growth underlying radiotherapy has not been studied. In the present study, tumor growth in tumor-bearing mice receiving radiation therapy with CA treatment was delayed compared to the control group and did not show any toxicity. Therefore, as a potential therapeutic approach to mitigate radiation-induced toxicity, CA might be used in tumor radiotherapy to improve the prognosis.

## 5. Conclusions

We found that CA attenuated radiation-induced endothelial dysfunction in vitro, including proliferation, senescence, and tube formation activity. We have also shown therapeutic effects of CA on radiation-induced enteritis, with the recovery of endothelial and epithelial dysfunction, focusing on the crosstalk between endothelial cells and epithelial cells. In particular, we identified EGF, a key factor secreted by endothelial cells to repair radiation-induced epithelial barrier dysfunction. Furthermore, by using a neutralizing anti-EGF antibody, we have shown the failure of the restoration of the radiation-induced epithelial barrier dysfunction and the expression of the related molecules in Caco-2 monolayers. The CM of CA-treated HUVECs or rEGF was administrated to a mouse model, and the results show recovery of radiation-induced epithelial dysfunction, including increased expression of epithelial barrier-related molecules. CA had no radioprotective effects in vivo during the radiotherapy for colorectal cancer, indicating CA treatment in radiotherapy could also be used for safe clinical applications. Thus, our study results suggest the use of CA as an effective radio-mitigator against radiation-induced enteritis.

## Figures and Tables

**Figure 1 cells-11-02544-f001:**
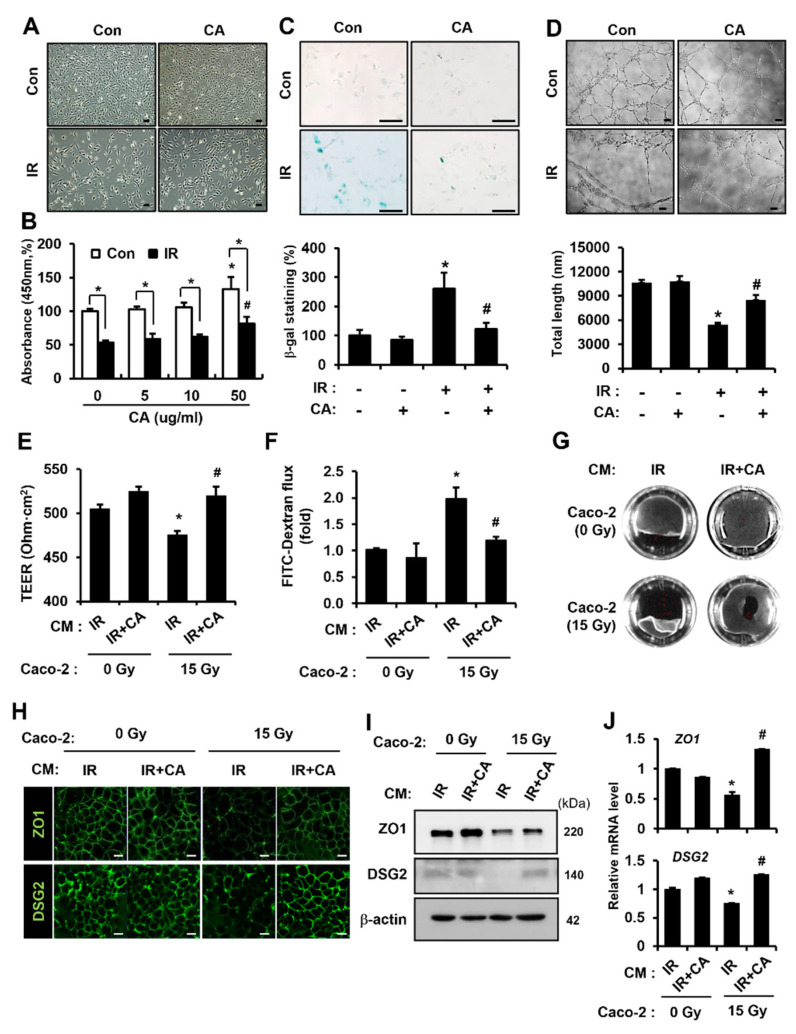
*Centella asiatica*-treated endothelial cells are recovered from radiation-induced dysfunction and protect against epithelial barrier damage. (**A**) Morphology and (**B**) viability of HUVECs after irradiation and treatment with *Centella asiatica* (CA). The effect of CA on HUVEC viability was assessed by CCK-8 assay. Bars represent the percentage of survival cells normalized to that of the corresponding control. Scale bars represent 100 μm. (**C**) Senescent activity of CA. Senescent HUVECs were quantified by a b-galactosidase assay. Senescent HUVECs were quantified and plotted as a bar graph (right). (**D**) Angiogenic activity assay. HUVECs were re-seeded onto Matrigel-coated wells in the presence or absence of CA. Total segments length per five fields were quantified and plotted as a bar graph (right). (**E**) The transepithelial electrical resistance (TEER) value of Caco-2 monolayers. The groups are as follows: conditioned medium (CM) of irradiated (IR)-HUVECs treating non-irradiated Caco-2 monolayers, CM of CA-treated irradiated (IR + CA)-HUVECS treating non-irradiated Caco-2 monolayers, CM of IR treating irradiated Caco-2 monolayers, and CM of IR + CA treating irradiated Caco-2 monolayers. (**F**) The flux of FITC-dextran (4 kDa) in the lower-chamber and (**G**) Dispase-based dissociation activity of each Caco-2 monolayer. (**H**) The intensity of zonula occludens 1 (ZO1) and desmoglein 2 (DSG2) on intercellular junction of Caco-2 monolayers and (**I**) protein levels of ZO1 and DSG2. (**J**) mRNA levels of *ZO1* and *DSG2* were assessed by qRT-PCR. Data are presented as the mean ± standard deviation of the mean; *n* = 3 per group. * *p* < 0.05 compared to CM of IR treating non-irradiated Caco-2 monolayers; # *p* < 0.05 compared to the CM of IR treating irradiated Caco-2 monolayers. Scale bars represent 10 μm. All data represent at least two independent experiments.

**Figure 2 cells-11-02544-f002:**
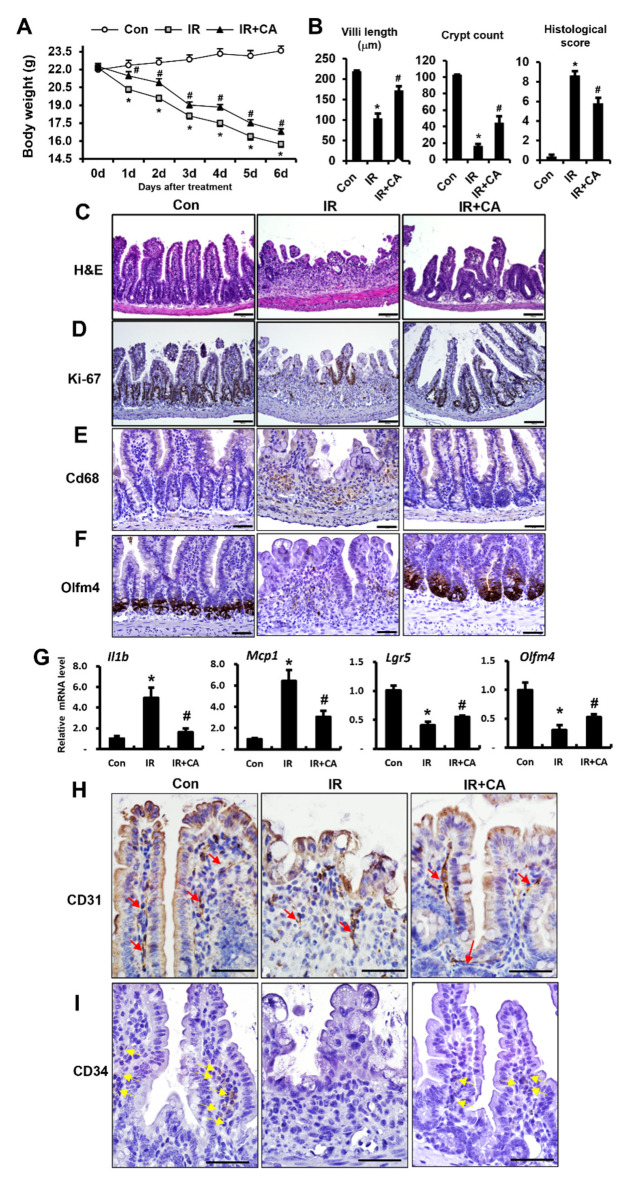
*Centella asiatica* alleviates radiation-induced enteritis in mouse model. (**A**) Body weights of control (Con), irradiated (IR), and *Centella asiatica* (CA)-treated irradiated mice (IR + CA). (**B**) The length of villi, crypts counting, and histological scoring in the small intestine were quantified. Histological scoring was assessed based on the degree of epithelial architecture maintenance, crypt disruption, vascular enlargement, and infiltration of inflammatory cells in the lamina propria of the ileum of Con, IR, and IR + CA mice groups (0 = none, 1 = mild, 2 = moderate, and 3 = high). (**C**) Hematoxylin and eosin (H&E) staining was performed using small intestine of Con, IR, and IR + CA groups. (**D**) Representative images of the small intestine stained with Ki-67 showing proliferation. Scale bars represent 100 μm. Representative images of the small intestine stained with (**E**) Cd68 and (**F**) Olfm4. Scale bars represent 50 μm. (**G**) qRT-PCR analysis demonstrating the mRNA levels of *Il1b*, *Mcp1*, *Lgr5*, and *Olfm4* in small intestine of each group. Representative images of the small intestine stained with (**H**) CD31 showing the positive endothelial cells and (**I**) CD34 showing the positive endothelial progenitor cells. Red arrows indicate the CD31-positive endothelial cells. Yellow arrows indicate the CD34-positive endothelial progenitor cells. Scale bars represent 50 μm. Data are presented as the mean ± standard error of the mean; *n* = 6 mice per group. * *p* < 0.05 compared to the Con group; # *p* < 0.05 compared to the IR group. All data represent at least two independent experiments.

**Figure 3 cells-11-02544-f003:**
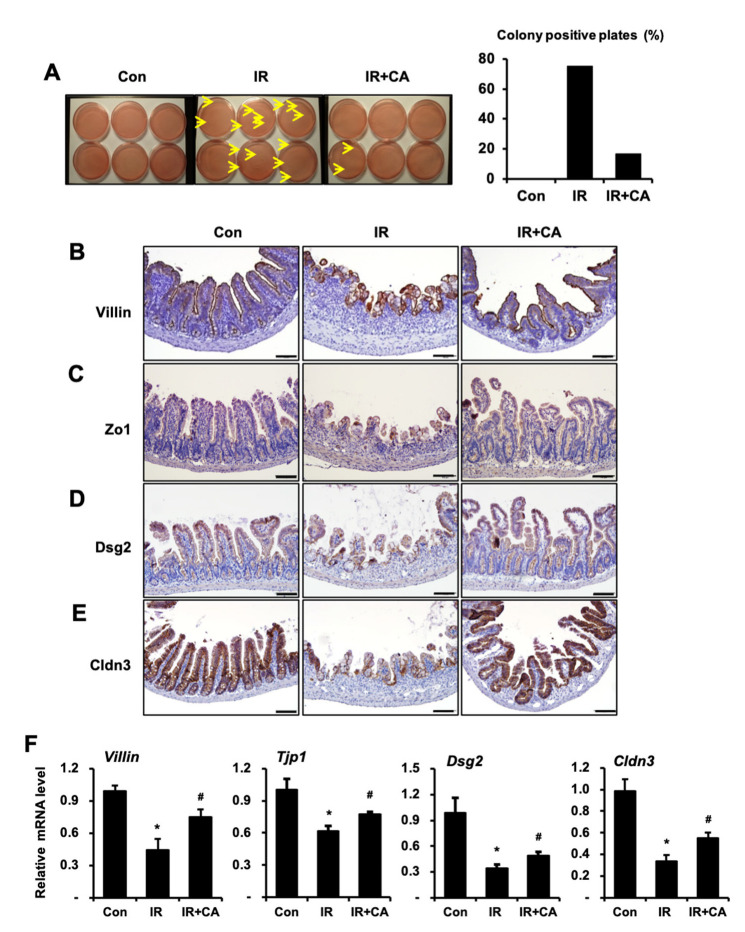
*Centella asiatica* ameliorates radiation-induced intestinal barrier dysfunction in vivo. (**A**) The bacterial colonies from mesenteric lymph node of control (Con), irradiated (IR), and *Centella asiatica* (CA)-treated IR group (IR + CA) were quantified. The graph is depicted as the percentage of individual mice exhibiting colonies in the group. *n* = 6 mice per group. (**B**–**E**) Representative images of small intestine stained with villin, zonula occludens 1 (Zo1), desmoglein 2 (Dsg2), and claudin 3 (Cldn3). (**F**) qRT-PCR analysis demonstrating the mRNA levels of *Villin*, *Tjp1*, *Dsg2*, and *Cldn3* in small intestine of each group. Data are presented as the mean ± standard error of the mean; *n* = 6 mice per group. * *p* < 0.05 compared to the Con group; # *p* < 0.05 compared to the IR group. Scale bars represent 100 μm. All data represent at least two independent experiments.

**Figure 4 cells-11-02544-f004:**
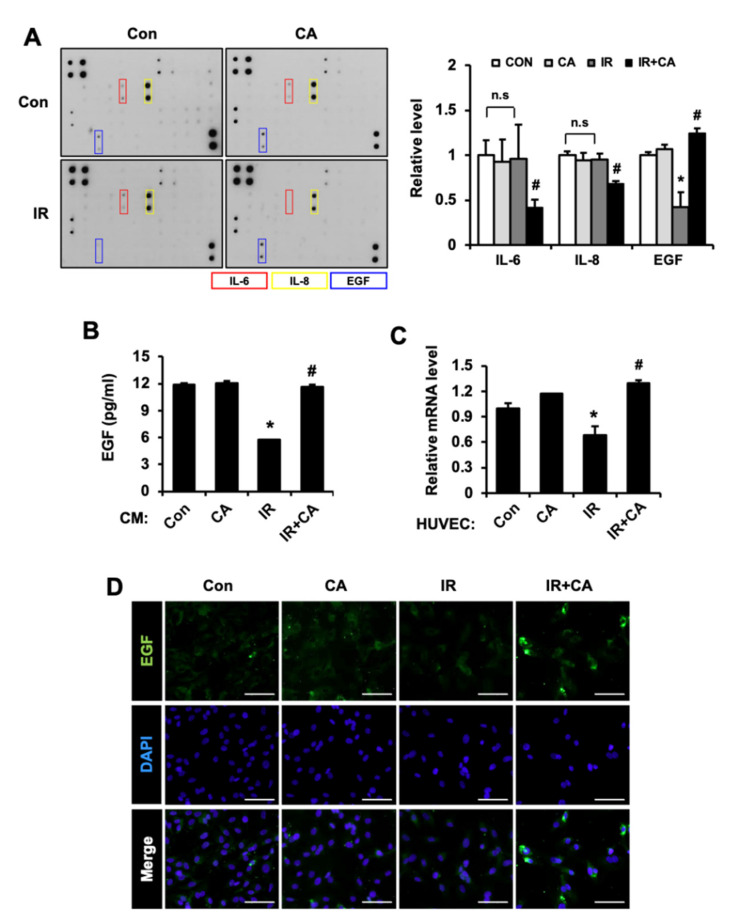
*Centella asiatica* accelerates epidermal growth factor production in irradiated endothelial cells. (**A**) The cytokine array was performed using the conditioned media of control (Con), irradiated (IR), CA-treated HUVECs (CA), and CA-treated irradiated HUVECs (IR + CA). The bar graph is shown as relative folds of interleukin (IL)-6, IL-8, and epidermal growth factor (EGF). (**B**) Secretion of EGF was quantified by ELISA. (**C**) The mRNA level of *EGF* was determined by qRT-PCR. (**D**) The EGF-positive cells were observed by confocal laser scanning microscope. Data are presented as the mean ± standard deviation of the mean; *n* = 3, * *p* < 0.05 compared to the Con group; # *p* < 0.05 compared to the IR group. Scale bars represent 50 μm. All data represent at least two independent experiments.

**Figure 5 cells-11-02544-f005:**
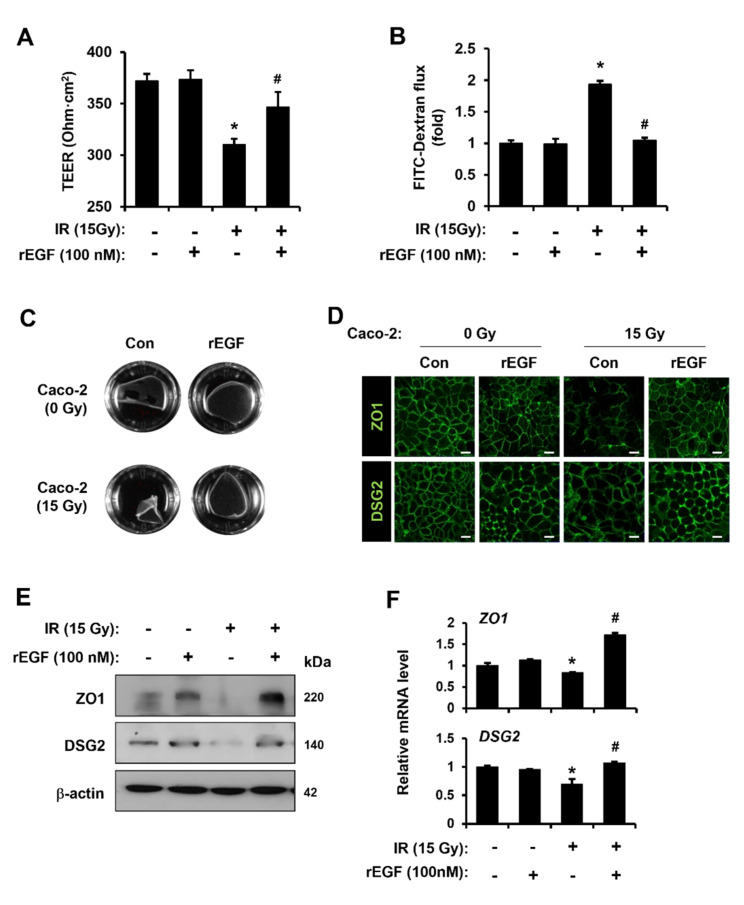
Epidermal growth factor restores radiation-induced epithelium barrier dysfunction with upregulation of ZO1 and DSG2. (**A**) Transepithelial electrical resistance (TEER) values of Caco-2 monolayers were determined after treatment with recombinant epidermal growth factor (rEGF; 100 nM). (**B**) The flux of FITC-dextran was measured using a microplate fluorescence reader (excitation at 450 nm and emission at 520 nm). The graph is shown as a fold of fluorescence normalized to control group. (**C**) The activity of dispase-based dissociation in Caco-2 monolayers with or without rEGF was observed. (**D**) The zonula occludens 1 (ZO1) and desmoglein 2 (DSG2) intensities of Caco-2 monolayers were observed using a confocal laser scanning microscope. (**E**) Protein levels of ZO1 and DSG2 were determined by western blot analysis. (**F**) mRNA levels of *ZO1* and *DSG2* were measured by qRT-PCR. Data are presented as the mean ± standard deviation of the mean; *n* = 3, * *p* < 0.05 compared to the Con group; # *p* < 0.05 compared to the IR group. Scale bars represent 10 μm. All data represent at least two independent experiments.

**Figure 6 cells-11-02544-f006:**
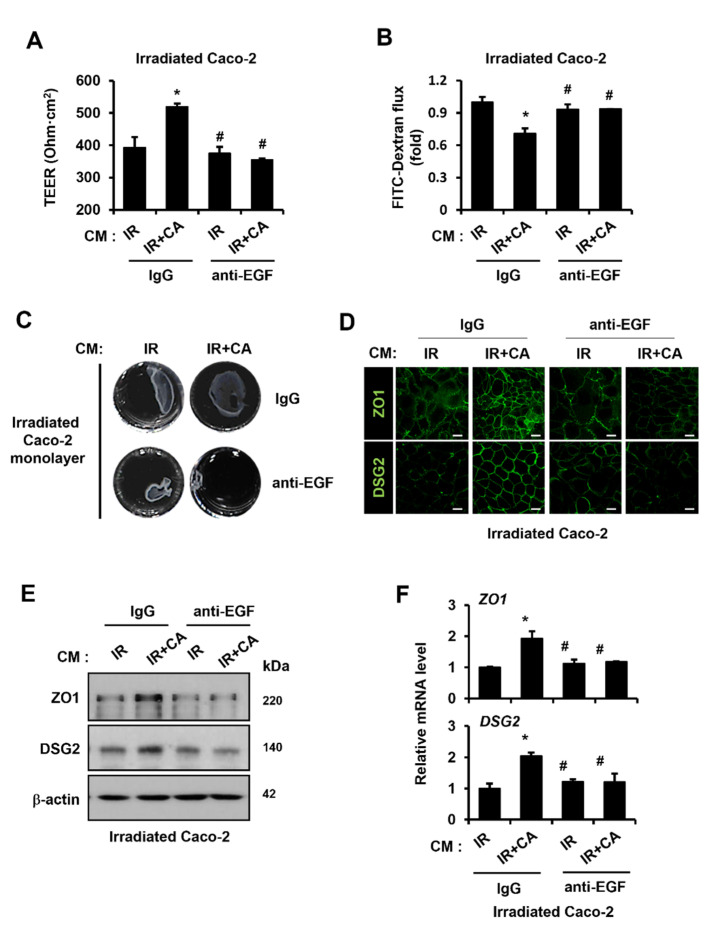
*Centella asiatica*-derived endothelial epidermal growth factor rescues radiation-induced barrier impairment with upregulation of zonula occludens 1 and desmoglein 2. (**A**) Transepithelial electrical resistance (TEER) value of irradiated Caco-2 monolayers on transwells was determined after treatment with CM of irradiated (IR) HUVECs or CA-treated IR (IR + CA) in presence of neutralizing antibody to EGF (anti-EGF, 100 ng/mL). The bar graph is shown as TEER value of each group. (**B**) The flux of FITC-dextran (4 kDa) in lower chambers was measured using a microplate fluorescence reader (excitation at 450 nm and emission at 520 nm). The bar graph is shown as a fold of flux of fluorescence normalized to irradiated Caco-2 monolayers treated with CM of IR. (**C**) Dispase-based dissociation activity of each irradiated Caco-2 monolayer was determined. Prepared irradiated Caco-2 monolayers were incubated in dispase II (2.4 U/mL) and collagenase type I for 30 min. After applying mechanical stress, the fragmentation of Caco-2 monolayers was observed using a digital camera. (**D**) The intensity of zonula occludens 1 (ZO1) and desmoglein 2 (DSG2) on the intercellular junction of irradiated Caco-2 monolayers was assessed by confocal staining. (**E**) Protein levels of ZO1 and DSG2 were assessed by western blot analysis. (**F**) mRNA levels of *ZO1* and *DSG2* were assessed by qRT-PCR. Data are presented as the mean ± standard deviation of the mean; *n* = 3 per group. * *p* < 0.05 compared to the CM of IR-treated irradiated Caco-2 monolayers; # *p* < 0.05 compared to the CM of IR + CA treated irradiated Caco-2 monolayers. Scale bars represent 10 μm. All data represent at least two independent experiments.

**Figure 7 cells-11-02544-f007:**
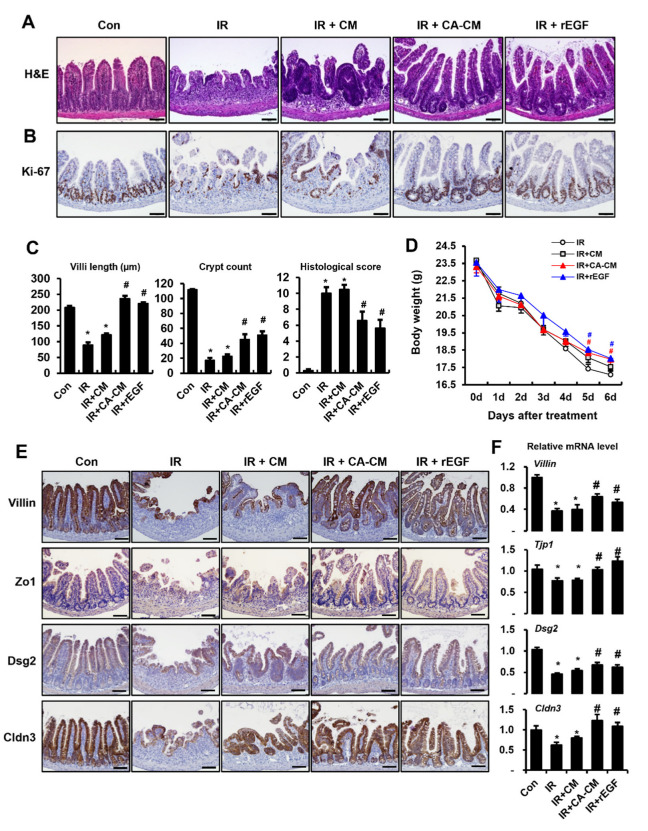
*Centella asiatica*-derived endothelial epidermal growth factor mitigates radiation-induced enteritis with epithelial barrier restoration in mouse model. Mouse groups are as follows: control (Con), irradiated (IR), irradiated mouse administrated with the conditioned media (CM) of irradiated HUVECs (IR + CM), CM of *Centella asiatica* (CA)-treated irradiated HUVECs (IR + CA-CM), and recombinant EGF (IR + rEGF). (**A**) Hematoxylin and eosin staining of mouse intestinal tissue sections was performed in each group. (**B**) Proliferation was assessed by staining intestinal sections with ki-67. (**C**) The lengths of villi and crypts in the intestinal sections were quantified. Histological scoring was assessed based on the degree of epithelial architecture maintenance, crypt disruption, vascular enlargement, and infiltration of inflammatory cells in the lamina propria of the ileum of Con, IR, IR + CM, IR + CA-CM, and IR + rEGF groups (0 = none, 1 = mild, 2 = moderate, and 3 = high). (**D**) Body weights of Con, IR, IR + CM, IR + CA-CM, and IR + rEGF groups were determined. (**E**) Immunohistochemistry against epithelial barrier-related molecules, e.g., villin, zonula occuludens 1 (Zo1), desmoglein 2 (Dsg2), and claudin 3 (Cldn3), in the Con, IR, IR + CM, IR + CA-CM, and IR + rEGF groups was performed. (**F**) mRNA levels of the epithelial barrier-related molecules in Con, IR, IR + CM, IR + CA-CM, and IR + rEGF groups were assessed by qRT-PCR. Data are presented as the mean ± standard error of the mean; *n* = 6 mice per group. * *p* < 0.05 compared to the Con group; # *p* < 0.05 compared to the IR group. Scale bars represent 100 μm. All data represent at least two independent experiments.

**Figure 8 cells-11-02544-f008:**
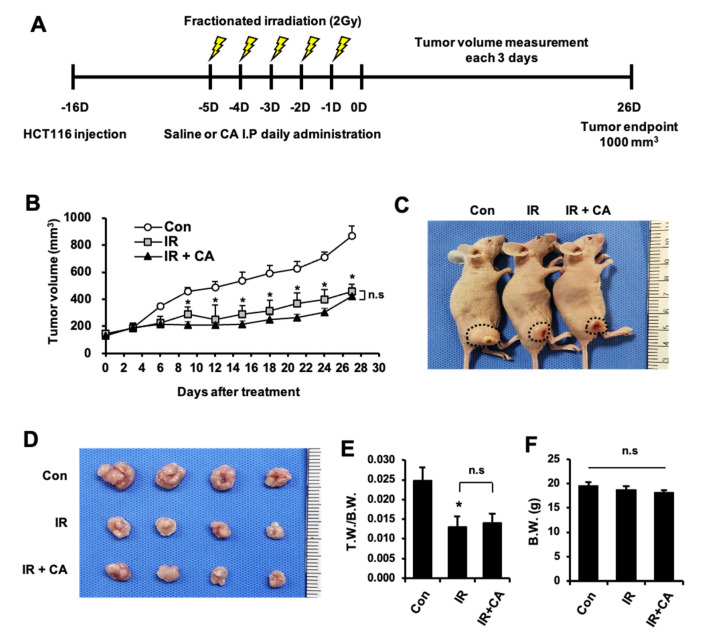
*Centella asiatica* treatment does not occur the radioprotective effect on colorectal tumor growth. Groups of tumor-bearing are as follows: control (Con), local irradiated (IR), local irradiated mouse administrated with *Centella asiatica* (IR + CA). (**A**) Scheme of the experimental procedure. Tumor-bearing mice were irradiated with g-rays (2 Gy × 5 fractionations for 5 days) followed by CA administration. Treatments began when tumor volume reached 100 mm^3^. (**B**) Tumor growth curves of the HCT-116 xenograft model after IR or IR plus CA treatment were determined. Representative images of (**C**) the tumor-bearing mice and (**D**) the excised tumor mass. (**E**) Tumor weight (T.W.) and body weight (B.W.) ratio was examined after mice were sacrificed. (**F**) B.W. of mice were examined. Data are presented as the mean ± standard error of the mean; *n* = 7 mice per group. * *p* < 0.05 compared to the Con group. “n.s” indicates not significant (*p* > 0.05). All data represent at least two independent experiments.

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
