# Peer review of "Centella asiatica-Derived Endothelial Paracrine Restores Epithelial Barrier Dysfunction in Radiation-Induced Enteritis"

_cells, 2022, doi:10.3390/cells11162544_

Round 1

Reviewer 1 Report

This is an interesting study demonstrating in vitro the beneficial effect of CA on radiation-induced dysfunction of endothelial cells (viability, proliferation, senescence and tube formation). EGF was identified, secreted by CA-treated endothelial cells in vitro, responsible for beneficial effects of irradiated endothelial cells’CM on epithelial cells. The beneficial effect of CA (and EGF) was confirmed in vivo on whole abdominal 13,5 Gy-irradiated mice.

Main comments

1-      About the main message of the manuscript: this is a well-written study with clearly exposed results despite numerous experimental conditions, the story is easy to follow. Nevertheless, I have some concerns about part of the conclusions. I am strongly convinced that CA has beneficial effects on radiation-induced small intestinal damage, however to my sense the fact that this is related to crosstalk between endothelial cells and epithelial cells has not been demonstrated in vivo. In vitro data are clear, but how can we be sure that CA-derived EGF is secreted by endothelial cells in vivo and that the beneficial effect on mice is due to endothelial cells’influence on epithelial cells ? To make this conclusion, study would have to demonstrate endothelial cells protection by CA treatment in vivo, as shown in vitro. For example, data have been published on the role of endothelial cells apoptosis in such radiation-induced GI toxicity by Paris et al. (2001), cited in the introduction section. Maybe CA could protect endothelial cells from radiation-induced apoptosis? Figure 2H shows better microvascular structure in intestinal villi of CA-treated animals, however, could this result from direct epithelial cells protection, keeping healthy epithelium and associated microvasculature? What about a direct effect of CA on epithelial cells in vivo? These reserves may be evoked in the discussion section.

2-      Introduction part: lines 34 and 35, please make a difference between accidental exposure and radiotherapy-associated toxicity. References 1 and 2 are not adapted, please find maybe some revues on the mechanisms of radiation-induced GI damage. Reference 3 concerns only RT-associated toxicity, a reference on accidental situations is lacking. Line 44: “communication…well-accepted” please add a reference. References 13-15 are not adapted, they are dealing with IDB, or just add in the sentence that strategies are promising against IBD and may be interesting in the context of radiation-induced GI damage.

3-      Materials and methods: line 119-120: after exposure, what does it means? The same day just after? Please precise. Line 134: for tube formation quantification, did authors look at branching points and loop numbers? Line 143: I do not understand “the ileum as a reference”. Please explain. Please also precise how many villi/crypts were measured/counted per slide. At this time point, GI damage is often inhomogeneous, with areas of severe epithelial loss closed to areas showing crypt regeneration. How was this taken into account to measure villus height and crypt numbers? Gene expression: was gene expression measured on the whole intestinal wall or after smooth muscles removing?

4-      Results: I do not understand the rationale for references 29-31 in the context of epithelium exposed to soluble factors secreted by surrounding cells. Figure 7A: the picture IR+CM shows numerous regenerating crypts. This does not correspond to the corresponding Ki67-stained section shown just below. Please explain.

5-      Discussion: please refer to first general comments to include reserves in the discussion section. Line 567/568 as in the introduction section, those symptoms more refer to as accidental exposure, please make a difference between accident and RT-associated symptoms (accelerated transit, chronic diarrhea, malabsorption, obstruction). Reference 39 refers only to RT-associated toxicity. Line 640: authors should evoke the interest to test orthotopic tumor model, in particular if effect may be driven by the vascular system.

6-      Conclusion: line 650: given the data obtained in vivo, the therapeutic effects of CA was associated with recovery of epithelial dysfunction, not endothelial (see comments 1).

Minor comments:

2.14 section: title must be in italic format and the text not.

Figure legends: errors in typing “m” instead of “µm” lines 318, 368, 371, 397, 457, 491, 535.

Author Response

#Reviewer 1

This is an interesting study demonstrating in vitro the beneficial effect of CA on radiation-induced dysfunction of endothelial cells (viability, proliferation, senescence and tube formation). EGF was identified, secreted by CA-treated endothelial cells in vitro, responsible for beneficial effects of irradiated endothelial cells’CM on epithelial cells. The beneficial effect of CA (and EGF) was confirmed in vivo on whole abdominal 13,5 Gy-irradiated mice.

Main comments

  1. About the main message of the manuscript: this is a well-written study with clearly exposed results despite numerous experimental conditions, the story is easy to follow. Nevertheless, I have some concerns about part of the conclusions. I am strongly convinced that CA has beneficial effects on radiation-induced small intestinal damage, however to my sense the fact that this is related to crosstalk between endothelial cells and epithelial cells has not been demonstrated in vivo. In vitro data are clear, but how can we be sure that CA-derived EGF is secreted by endothelial cells in vivo and that the beneficial effect on mice is due to endothelial cells’influence on epithelial cells ? To make this conclusion, study would have to demonstrate endothelial cells protection by CA treatment in vivo, as shown in vitro. For example, data have been published on the role of endothelial cells apoptosis in such radiation-induced GI toxicity by Paris et al. (2001), cited in the introduction section. Maybe CA could protect endothelial cells from radiation-induced apoptosis? Figure 2H shows better microvascular structure in intestinal villi of CA-treated animals, however, could this result from direct epithelial cells protection, keeping healthy epithelium and associated microvasculature? What about a direct effect of CA on epithelial cells in vivo? These reserves may be evoked in the discussion section.
  • Thank you for your comments. To respond to the comments, we performed additional experiments (CD34 stain in Figure 2I). In radiation-induced intestinal injury, endothelial cell is sensitively damaged and a critical factors for protection and mitigation against to radiation exposure. We also identified that CA could not improve the survival rate in an epithelial cell line in 4 Gy radiation exposure. In addition, there is a little information about the communication of endothelial cells and epithelial cells. in this study, we focused on the crosstalk between endothelial cells and epithelial barrier by CA.
  1. Introduction part: lines 34 and 35, please make a difference between accidental exposure and radiotherapy-associated toxicity. References 1 and 2 are not adapted, please find maybe some revues on the mechanisms of radiation-induced GI damage. Reference 3 concerns only RT-associated toxicity, a reference on accidental situations is lacking. Line 44: “communication…well-accepted” please add a reference. References 13-15 are not adapted, they are dealing with IDB, or just add in the sentence that strategies are promising against IBD and may be interesting in the context of radiation-induced GI damage.
  • To respond to the comments, we revised and added references.
  1. Materials and methods: line 119-120: after exposure, what does it means? The same day just after? Please precise. Line 134: for tube formation quantification, did authors look at branching points and loop numbers? Line 143: I do not understand “the ileum as a reference”. Please explain. Please also precise how many villi/crypts were measured/counted per slide. At this time point, GI damage is often inhomogeneous, with areas of severe epithelial loss closed to areas showing crypt regeneration. How was this taken into account to measure villus height and crypt numbers? Gene expression: was gene expression measured on the whole intestinal wall or after smooth muscles removing?
  • To respond to the comments, we revised and added the sentences.
  1. Results: I do not understand the rationale for references 29-31 in the context of epithelium exposed to soluble factors secreted by surrounding cells. Figure 7A: the picture IR+CM shows numerous regenerating crypts. This does not correspond to the corresponding Ki67-stained section shown just below. Please explain.
  • To respond to the comments, we revised the sentences. And we indicated the ki-67 positive regenerative crypts with red arrow in the IR+CM group. The results corresponded to the H&E stain.    

  1. Discussion: please refer to first general comments to include reserves in the discussion section. Line 567/568 as in the introduction section, those symptoms more refer to as accidental exposure, please make a difference between accident and RT-associated symptoms (accelerated transit, chronic diarrhea, malabsorption, obstruction). Reference 39 refers only to RT-associated toxicity. Line 640: authors should evoke the interest to test orthotopic tumor model, in particular if effect may be driven by the vascular system.
  • Thanks for your comments. To respond to the comments, we revised the sentences and the references. In vascular change in xenograft cancer model, we agree your comments. in histological analysis, we did not find the alteration of vascular system between IR+CA group and IR group (Data not shown).
  1. Conclusion: line 650: given the data obtained in vivo, the therapeutic effects of CA was associated with recovery of epithelial dysfunction, not endothelial (see comments 1).
  • Thanks for your comments. To respond to the comments, we added the sentences. We also firstly identified the effects of CA in the irradiated epithelial cells. But in high dose irradiation (over 4Gy), CA did not show the proliferative effects directly. Therefore, we focused on the communication of endothelial cells and epithelial cells in CA treatment. To identify the effects of CA on the irradiated endothelial cells in vivo model, we performed immune-stain of CD34 in the mouse model.

Minor comments:

  1. 14 section: title must be in italic format and the text not.
  • To respond to the comments, we revised the title in the Materials and Methods section.
  1. Figure legends: errors in typing “m” instead of “µm” lines 318, 368, 371, 397, 457, 491, 535.
  • To respond to the comments, we revised the words.

Reviewer 2 Report

Kwak et al. have submitted a manuscript entitled “Centella asiatica-derived endothelial paracrine restores epithelial barrier dysfunction in radiation-induced enteritis” for publication in Cells.

The authors have analyzed the effects of conditioned media from Centella asiatica-treated irradiated HUVEC cells on radiation-induced epithelial barrier damage. They also report that treatment with EGF showed beneficial effect, and report an effect also in an enteritis mouse model, but not in colon cancer.

The following points are recommended to strengthen the study.

1. The authors report a radiation-induced decrease of TEER in Caco-2 cells, and effects on mouse intestine. Whereas in mouse one barrier-forming tight junction protein was detected (claudin-3), only scaffolding ZO-1 and desmosome dsg2 was detected in Caco2. For proof of concept and for better interpretation of functional barrier data, tight junction membrane proteins should be analyzed in both models. 

2. Intestinal epithelial tissues appear rather destroyed after IR. Did the authors perform a morphometric analysis for better comparison of epithelial surface area? Considering surface correction, an even higher expression of proteins in epithelial cells might be detectable, as epithelial cell number looks markedly reduced after IR.

3. The authors used conditioned medium from irradiated HUVECs. Did the authors also use conditioned medium of control HUVECs, for proof of concept?

4. Some descriptions appear rather uncommon and correction of the manuscript by a native English-speaking colleague regarding English language and style would be recommended. The “Ohmn” values should be transformed into ohm x cm2.

Author Response

#Reviewer 2

Kwak et al. have submitted a manuscript entitled “Centella asiatica-derived endothelial paracrine restores epithelial barrier dysfunction in radiation-induced enteritis” for publication in Cells.

The authors have analyzed the effects of conditioned media from Centella asiatica-treated irradiated HUVEC cells on radiation-induced epithelial barrier damage. They also report that treatment with EGF showed beneficial effect, and report an effect also in an enteritis mouse model, but not in colon cancer.

The following points are recommended to strengthen the study.

  1. The authors report a radiation-induced decrease of TEER in Caco-2 cells, and effects on mouse intestine. Whereas in mouse one barrier-forming tight junction protein was detected (claudin-3), only scaffolding ZO-1 and desmosome dsg2 was detected in Caco2. For proof of concept and for better interpretation of functional barrier data, tight junction membrane proteins should be analyzed in both models. 
  • Thank you for the comments. We also identified the expression of claudin 3 in the irradiated Caco-2 monolayers. Unfortunately, there was no difference in the claudin 3 expression between IR and IR+CA. Claudin3 is a sensitive tight junction by irradiation condition in the small intestine. Otherwise, the Caco-2 cells are characterized by colorectal epithelial cells. We speculate that there is a different expression of claudin3 between small intestine and large intestine.  
  1. Intestinal epithelial tissues appear rather destroyed after IR. Did the authors perform a morphometric analysis for better comparison of epithelial surface area? Considering surface correction, an even higher expression of proteins in epithelial cells might be detectable, as epithelial cell number looks markedly reduced after IR.
  • Thank you for the comments. In 6 days after irradiation, the epithelium is severely damaged with detachment, delayed proliferation, and destroyed epithelial integrity. Otherwise, the endothelial cells is sensitive to the radiation exposure and are occurred apoptosis, senescence.
  1. The authors used conditioned medium from irradiated HUVECs. Did the authors also use conditioned medium of control HUVECs, for proof of concept.
  • Previously, we identified CM of non-IR HUVEC improved epithelial barrier function compared with CM of IR HUVEC. And as shown in Fig 4a, the secretion of EGF increased in CM of non-IR HUVEC compared with CM of IR HUVEC.   
  1. Some descriptions appear rather uncommon and correction of the manuscript by a native English-speaking colleague regarding English language and style would be recommended. The “Ohmn” values should be transformed into ohm x cm2.
  • To respond to the comments, we revised the unit.

Reviewer 3 Report

The research article submitted by Kwak et al. aimed to investigate how CA as a radio-mitigator ameliorated IR-induced enteritis. They found CA treatment of irradiated endothelial cells induced the secretin of EGF, which is a critical factor for maintaining epithelial barrier. In general, this work is interesting and meaningful to develop the measures for alleviation of radiation-induced enteritis.

The manuscript was clearly written. But I have a few concerns and queries that may be considered before publication of the article.

1.      Please briefly introduce the molecules of ZO1 and DSG2 in the Introduction part.  

2.      Page 2, line 58: Indian should be India.

3.      The method “2.14. Dispase-based dissociation assay” was not described clearly.

4.      In Figure 7D, markers “#” were not presented clearly.

5.      I'm curious whether CA can also induce intestinal epithelium to secrete EGF under irradiation. It may be discussed.

Author Response

Reviewer 3

The research article submitted by Kwak et al. aimed to investigate how CA as a radio-mitigator ameliorated IR-induced enteritis. They found CA treatment of irradiated endothelial cells induced the secretin of EGF, which is a critical factor for maintaining epithelial barrier. In general, this work is interesting and meaningful to develop the measures for alleviation of radiation-induced enteritis.

The manuscript was clearly written. But I have a few concerns and queries that may be considered before publication of the article.

  1. Please briefly introduce the molecules of ZO1 and DSG2 in the Introduction part.  

- Thank you for your comments. To respond to the comments, we added the sentence.

  1. Page 2, line 58: Indian should be India.
  • To respond to the comments, we revised the word.
  1. The method “2.14. Dispase-based dissociation assay” was not described clearly.
  • To respond to the comments, we added the sentence.
  1. In Figure 7D, markers “#” were not presented clearly.
  • To respond to the comments, we revised Figure 7D.
  1. I'm curious whether CA can also induce intestinal epithelium to secrete EGF under irradiation. It may be discussed.
  • Thank you for your comments. To respond to the comments, we added the sentence. Previously, we also supposed the direct effects of CA on irradiated epithelial cells. Unfortunately, CA could not improve the survival rate in an epithelial cell line in 4 Gy radiation exposure. Therefore, in high dose irradiated condition, CA directly affects damaged endothelial cells rather than epithelial cells.

Round 2

Reviewer 2 Report

The authors have adressed all points, but moderate language editing is still required, especially of the revised/added text.